# EUPollMap: The European atlas of contemporary pollen distribution maps derived from an integrated Kriging interpolation approach

Fabio Oriani[1,2], Gregoire Mariethoz[1], and Manuel Chevalier[3,1]

[1]Faculty of Geosciences and Environment, University of Lausanne, CH-1015 Lausanne, Switzerland
[2]Agroscope, Remote Sensing Team, Division Agroecology and Environment, CH-8046 Zürich, Switzerland
[3]Institute of Geosciences, Sect. Meteorology, Rheinische Friedrich – Wilhelms - Universität Bonn, Auf dem Hügel 20, 53121 Bonn, Germany

**Correspondence:** Fabio Oriani (fabio.oriani@protonmail.com)

**Abstract.** Modern and fossil pollen data are widely used in paleoenvironmental research to characterise past environmental changes in a given location. However, their discrete and discontinuous nature can limit the inferences that can be made from them. Deriving continuous spatial maps of the pollen presence from point-based datasets would enable more robust regional characterization of such past changes. To address this problem, we propose a comprehensive collection of European pollen presence maps including 194 pollen taxa derived from the interpolation of pollen data from the Eurasian Modern Pollen Database (EMPD v2) restricted to the Euro-Mediterranean Basin. To do so, we developed an automatic Kriging-based interpolation workflow to select an optimal geostatistical model describing the spatial variability for each taxon. The output of the interpolation model consists in a series of multivariate predictive maps of Europe at 25-km scale, showing the occurrence probability of pollen taxa, the predicted presence based on diverse probability thresholds, and the interpolation uncertainty for each taxon. Combined visual inspections of the maps and systematic cross-validation tests demonstrated that the ensemble of predictions is reliable even in data-scarce regions, with a relatively low uncertainty, and robust to complex and non-stationary pollen distributions. The maps, freely distributed as GeoTIFF files, are proposed as a ready-to-use tool for spatial paleoenvironmental characterization. Since the interpolation model only uses the coordinates of the observation to spatialise the data, the model can also be employed with fossil pollen records (or other presence/absence indicators), thus enabling the spatial characterization of past changes, and possibly, their subsequent use for quantitative paleoclimate reconstructions.

## 1 Introduction

Fossil pollen data are commonly used to document how different environments responded to past global climate forcing and events (Bartlein et al.; Dallmeyer et al., 2022). In particular, continental-scale studies of past land cover changes or biome data-model comparisons are particularly informative because they allow extracting common trends from large datasets (Gaillard et al.; Trondman et al.; Zanon et al., 2018; Githumbi et al.). Pollen data are also commonly employed to quantitatively reconstruct past climate using statistical models built on pollen-climate relationships derived from modern pollen observations (Birks et al., 2010; Chevalier et al., 2020). These reconstructions have been instrumental to improving our understanding of

past climate dynamics at various timescales (Kaufman et al., 2020; Herzschuh et al., 2023a, b; Marsicek et al., 2018; Routson et al., 2019) and to evaluate Earth System Model (ESM) simulations over land (Liu et al., 2014; Mauri et al., 2014; Weitzel et al., 2019).

Despite the recent accumulation of thousands of fossil pollen records in public repositories (Herzschuh et al., 2022; Williams et al., 2018), most pollen-based analyses remain performed at the site level. Each pollen sample is analysed in isolation from the others, and the results (*e.g.* land cover estimates or climate reconstructions) are then merged together to extract a regional/continental signal (Marsicek et al., 2018; Mauri et al., 2015; Herzschuh et al., 2023b). This approach is imperfect because it generates coarse spatial representations with abrupt local transitions relative to the regional grouping of the samples. Moreover, it generally does not take into account the local data density and variability, and assessing the spatial uncertainty of the estimations is generally not possible.

Advanced geostatistical techniques allow drawing spatial information based on data density, presence, and spatial variations. However, these techniques require a minimum density and quality in the studied data to produce reliable estimates. The recent growth of harmonised modern (Davis et al., 2020; Whitmore et al., 2005) and fossil (Williams et al., 2018; Herzschuh et al., 2022) pollen data now allow for the use of such techniques, which represent, as such, an interesting step forward in the analysis of large scale compilations of pollen data. In particular, the spatial covariance, a fundamental function of geostatistics, can be robustly estimated from pollen data in Europe or North America where hundreds of records are available. While the quantity of pollen grain observed for a given pollen taxon is affected by many processes (see for instance Chevalier et al. (2020)) and is, therefore, heterogeneous over space, its presence (*i.e.* the observation of one or more grains at any location) is subject to lower complexity. As such, binary presence data can be interpolated into space more robustly, irrespective of the characteristics of the sampling environment.

However, estimating the complex distribution maps of pollen occurrence probability across large regions and for many taxa is difficult. While the commonly used polynomial interpolation techniques could be an obvious solution, imposing an arbitrary spatial model (e.g. linear, quadratic, or cubic) and ignoring the uneven spatial distribution of the pollen samples limits the accuracy and reliability of such interpolations. Therefore, we developed a model based on Kriging (Matheron, 1963; Chiles and Delfiner, 2009) to spatialize our point-based observations and estimate occurrence probability maps. Kriging was preferred over other types of spatial interpolation techniques because it is based on a spatial model inferred from the observations and it minimizes the local bias and error variance. To enable its use over a large number of taxa, we embedded it in an automatic framework that preprocesses the pollen data, chooses the best type of spatial model for each pollen taxon, and generates the interpolated maps.

The goal of the present study is thus to realize, for the first time to our knowledge, a collection of raster maps representing the probability of occurrence of 194 pollen taxa observed across the Euro-Mediterranean Basin, compiled in an atlas called EUPollMap. For every taxon considered in EUPollMap, the output consists of a raster file with three layers showing: 1) the pollen occurrence probability, 2) the discrete occurrence based on probability thresholds, and 3) the uncertainty of the predictions. The paper is organized along two main axes with section 2 that describes the automatised Kriging methodology we

developed, and section 3 that introduces the cartographic products of the atlas with visual examples and reliability assessments. We contextualise the value of the research in section 4.

## 2 Methods: Spatial interpolation of pollen-presence data

### 2.1 The Indicator Kriging method

Kriging is a standard geostatistical interpolation technique that was first formalized in the early '60s (Matheron, 1963) and used since in various fields of geosciences such as e.g. mineral resources (Goovaerts et al., 1997; Sadeghi et al., 2015), hydrogeology (Varouchakis and Hristopulos, 2013), or soil properties (Emery and Ortiz, 2007; Minasny and McBratney, 2016). Several comparative studies have shown that Kriging produces robust and regionally-smooth interpolations, while minimizing the error variance and bias at the interpolated locations (see e.g. Zimmerman et al., 1999; Wagner et al., 2012; Oriani et al., 2020). Kriging interpolates discrete data $Z$ by estimating the target variable at any location of a pre-defined region of interest as a weighted mean of nearby data values, with the weights being computed by the resolution of a system of equations based on the semivariogram function $\gamma$.

The semivariogram, which is at the core of the Kriging algorithm, is a function that quantifies the spatial variability of the observed data as a response to the distance among them. Given the spatial variable $Z(x)$, defined at spatial locations $x$, with the hypothesis of stationarity (*i.e.* assuming that the statistical properties of $Z$ are uniform in space), the experimental semivariogram for $Z$ is estimated from the observed data as $\hat{\gamma}(h) = E[(Z(x) - Z(x+h))^2]/2$, where $Z(x)$ and $Z(x+h)$ are any pair of observations of $Z$ at distance $h$ and $E[\cdot]$ is the average operator among all pairs of points with a similar $h$, grouped in discrete $h$ intervals (lags).

Then, a parametric semivariogram model $\gamma$ is chosen among a family of pre-determined positive-definite functions and fitted to the experimental semivariogram $\hat{\gamma}$ mostly using a least-square approach. Different types of models are possible depending on the structure of the data and, usually, the model that fits the best with the experimental semivariogram, either by manual fit or by minimizing the error, is used in the Kriging system. Here, we consider the exponential model (see Isaaks and Srivastava, 1989, p.374):

$$\hat{\gamma}(h) = 1 - \exp(\frac{-3h}{a}) \tag{1}$$

where $a$ is the range parameter explained below. Among the standard models used in Kriging, this one is recommended for Indicator Kriging (p.102 Chiles and Delfiner, 2012; Dubrule, 2017), which is the modeling strategy adopted in this study (see below in the current section). Examples of experimental and fitted semivariogram models are shown in the panels b of Figures 1, 2, and 3.

Once fitted, the semivariogram model can be interpreted as follow. If the modelled curve intersects the origin of the axes, it indicates that the variation among adjacent observations ($h = 0$) is close to zero, while intersecting the y-axis at values larger than zero indicates discontinuities in adjacent data. The shape of the model curve along the x-axis (increasing lag) describes the variability over larger distances, where a steep slope indicates sharp variations. Often the model curve reaches a plateau,

whose corresponding $h$ value, called the range ($a$ in equation 1), indicates the average correlation length of the spatial structure. Above the range, the variable values are on average not correlated. In addition, since long lags are usually not represented by many pairs of data points, it is common practice to limit the model fitting to the data below a fixed maximum lag threshold. In the atlas presented in this paper, this threshold is set to 3100 km, which corresponds to the 80-th percentile of the distribution of all pairwise distances between the observations.

Two types of Kriging are considered here: Ordinary Kriging (OK), which assumes $Z$ to be stationary with no regional trend, and Universal Kriging with external drift (UK), implying the existence of a regional trend, given or estimated. In this study, we performed preliminary tests comparing OK with UK using elevation as external drift for different taxa datasets (supplemental material 3). OK allowed including all data points with reasonable computation time, while UK required an excessive computational burden. Also, elevation did not correlate with the pollen presence in the analyzed data, so that its inclusion did not sensibly affect the prediction when used as external drift in the Kriging model. For these reasons OK was preferred over UK.

Finally, the observation data for $Z$ (in this study) are binary and can only take two values: the pollen taxon is observed ($Z = 1$) or not observed ($Z = 0$). OK is therefore applied as an Indicator Kriging interpolation. This way, the first Kriging output map for $Z$ is the expected value, varying continuously in space between 0 and 1, which can be interpreted as the probability of occurrence of the pollen taxon. Occasionally, the interpolated value can lie just outside this interval (*e.g.* it is expected in situations where the Kriging weights are negative). In such cases, the values are bounded to either 0 or 1. The second output map is the Kriging variance and indicates the uncertainty of the prediction depending on the data amount, their spatial distribution, and the semivariogram model (Goovaerts et al., 1997, p.179). Generally, the variance is lowest around data points and increases with distance. In this study, the Kriging system is solved for every pollen taxon separately, using the python package PyKrige (https://pypi.org/project/PyKrige/).

## 2.2 Automatic interpolation workflow

The Kriging technique is usually employed in a supervised context, where the semivariogram model and its parameters are adjusted by examining the experimental semivariogram plot and the interpolation results in a trial-and-error approach. This is necessary to avoid overfitting the model semivariogram to the data, which can lead to unrealistic interpolations. However, when many datasets have to be interpolated, as is the case with the European pollen taxa, supervising the model setup for each taxon in a objective and consistent way is not feasible.

For this reason, following previous contributions (Desassis and Renard, 2013), we developed an automatic python interpolation workflow for the choice and optimization of the semivariogram model. The model fitting was based on having well represented lags, at least until a prescribed maximum lag threshold (here defined as 3100km, see section 2.1). Monotonicity and a positive slope are expected in a semivariogram model, but in case of a noisy experimental semivariogram (which is the case for some of the observed pollen taxa), unconstrained fitting can lead to a negative slope in the model. For this reason, we imposed flat or monotonic-positive model functions.

The probability maps were generated for each pollen taxon with the following steps:

1. If the dataset presents all-0 (*i.e.* the taxon is not observed in the study area) or all-1 (*i.e.* the taxon is observed at every sampling location) data, generate a 0/1 field on the defined grid as output Kriging mean and a 0 field as output Kriging variance, then go to step 5.

2. Compute the experimental semivariogram and calibrate its model (see section 2.1). When the data have little to no spatial correlation, the fitted semivariogram model tends to become a constant function, which subsequently leads to constant estimated mean and variance fields.

3. Solve the Kriging prediction with the optimised model parameters at every location of the interpolation grid to obtain the mean and variance maps. A mask based on the coastal perimeter is used to exclude large water bodies.

4. Generate a discrete occurrence map by applying a series of fixed thresholds (0.2, 0.4, 0.5, 0.6, 0.8) on the mean map.

5. Export an ESRI GeoTIFF file of the georeferenced output maps (see the metadata, Table 1).

6. Export the pollen presence/absence dataset as an ESRI Shapefile.

## 2.3 Validation strategy

We constituted an example dataset by identifying a series of common species for Europe with characteristic spatial features (e.g., broad extent, rare species, discontinuous distributions). The interpolated probability surfaces and their variances were then visually inspected and compared with the original observations. Moreover, the reliability of all the species interpolations was assessed with reliability plots (Murphy and Winkler, 1977), which are common quantitative and graphical representations in geo- and atmospheric sciences (Bröcker and Smith, 2007; Allard et al., 2012). A reliability plot is generated by splitting the dataset in training and validation data and the probability of occurrence predicted by the model is compared with the occurrence frequency observed in the validation data. For example, for grid cells with a predicted probability of occurrence around 0.2, the occurrence frequency observed from the validation data should be close to 0.2 for the prediction to be reliable. The predicted probability range [0-1] is divided in ten discrete bins to group the validation locations and co-located occurrence data. Then the sample occurrence probability values are plotted against the observed frequency. If the plot points lie along the bisector (*i.e.* the 1:1 line), the predictions can be considered reliable.

The way binary presence/absence data aggregate in space determines the estimated probability of occurrence. Some values are rarely found in the output probability map, therefore a high amount of validation data is needed for the reliability plot to be representative of all probability bins (Jolliffe and Stephenson, 2012). Moreover, the sampling for these data cannot be stratified according to the posterior probability values, which are not available a priori. To cope with this limitation, we randomly removed 50% of the data to approach stable statistical values for all bins of the reliability plot.

## 3 Application: The European atlas of modern pollen distributions

### 3.1 Definition of the study area

Distances between grid points are central to Kriging. As such, we used the spatial CRS EPSG:3034 that better respects distances than the standard CRS WGS84 that severely distorts distances when large latitudinal ranges are covered.

The dataset is limited to data located inside and near the spatial interpolation grid chosen to define the maps (see metadata in Table 1). The distance limits for data outside the grid, for both the E-W and N-S borders, is defined as 5% of the total longitudinal length of the map (approximately 242 Km). This ensures that the borders of the grid are surrounded by data, where possible, to limit extrapolation biases near the edges of the studied domain.

### 3.2 Source data

#### 3.2.1 Modern pollen data

The pollen-presence point data used in this study belong to the Eurasian Modern Pollen Database (EMPD) v.2 (Chevalier et al., 2019; Davis et al., 2020), a community-based, open-source database including 8134 pollen samples from all over Eurasia and parts of North Africa. To develop and test the interpolation workflow, we restricted this dataset to Europe, where data density is the highest. The dataset is composed of a mix of sample types, including surface layers of lake and bog sediments, moss
polsters, peat, and other data sources in very low proportions. To avoid redundancy and simplify the classification, the name of the pollen types from EMPD2 were grouped into a lower taxonomic resolution level and aligned to the globally harmonised pollen taxonomy of Herzschuh et al. (2022).

    Determining the proper absence of a pollen taxon can only be done with extensive vegetation surveys, which is unpractical at the European scale. Moreover, such surveys cannot be done for fossil observations. Therefore, we chose to analyze the
170 EMPD2 dataset as we would analyze the fossil records. For taxa that produce large quantities of pollen grains (grasses, pines), low percentages usually represent long-distance transport to the surroundings of the collection site, without the actual taxon presence (Lisitsyna et al., 2011). Assuming that their non-observation is proof of absence is therefore reasonable. On the other hand, rare taxa or low-pollinating taxa are more difficult to observe in both modern and fossil settings. It is common to observe them in one sample and not in the neighboring one. Using Kriging at the regional target scale for this study, this problem is
175 mitigated since the presence is assessed as a continuous probability variable, computed as a weighted mean from multiple neighbor presence/absence data.

    The data were preprocessed as follows:

- The data coordinates are transformed from the coordinate reference system (CRS) EPSG:4326 "WGS84", used for global data, to EPSG:3034 "ETRS89-extended / LCC Europe", used for European data, to reduce the local deformation for the
studied domain. Records with missing or invalid coordinates are discarded.

- The considered taxa belong to the following categories: dwarf shrubs (DWAR), herbs (HERB), liana (LIAN), palms (PALM), succulents (SUCC), trees and shrubs (TRSH), and uplands herbs (UPHE).

- The pollen counts are binarized to indicate the presence (1) or absence (0) at every location. Based on the hypothesis that all the pollen types in a sample have been detected, we assume that any sample location where a taxon has not been observed corresponds to an absence datum for that taxon. This results in a consistent point dataset for all taxa.

- Samples with identical coordinates are merged. In such case, a taxon is considered present if it is observed in at least one of the samples, and absent if not.

- Following Herzschuh et al. (2022), the taxa are grouped into 194 consolidated pollen taxa names (see the taxa list in supplemental material 1). This includes pollen types which present redundant naming or are not distinguishable in the count. This allows a more consistent point-presence distribution.

### 3.2.2 Reference plant dataset

To evaluate our probabilistic forecasts of the pollen presence, we also assess how the spatialized pollen data compare with the modern atlas of European tree distributions of Mauri et al. (2017). For the most part, the tree presence data are derived from national forest-monitoring surveys and interpolated over 1-km regular grids. Unfortunately, the spatial extent of this dataset is more limited than the pollen one as it only covers western and central Europe. In addition, there are differences among the two datasets that pertain to both the nature of the data (plant vs pollen) and the type of data collections (intensive forest inventories vs discrete field sampling to collect pollen samples). Therefore, this pollen-plant distribution comparison only serves as a broad visual assessment of the ability of the model to capture the main vegetation distribution.

### 3.3 Structure of the EUPollMap atlas

The atlas is a collection of 194 multivariate maps representing the interpolated pollen presence probability across a geographic domain that covers Europe and its main islands, as well as the northern edge of Africa at a 25-km resolution (see Fig. 1). This resolution is a tradeoff between the data density and our goal to provide a spatialised representation of the pollen observations. For each taxon, the output data is a set of raster maps exported as a GeoTIFF file that includes 1) the pollen occurrence probability map, corresponding to the Kriging interpolation mean, 2) a map with the discrete probability of occurrence, and 3) the occurrence uncertainty map (Kriging variance) (see Table 1 and the examples below). Each map file is complemented with a georeferenced shapefile containing the preprocessed source dataset (Table 2) that can be imported in any GIS software. The shapefile includes 5362 data points that document the presence or absence status of the taxon with the attribute POLLEN_PRE. Every taxa folder of the atlas also contains a summary pdf file with the output maps, point data, and semivariogram model plot (similar to Figs. 1–3).

**Table 1.** Main metadata of the pollen presence maps.

| Raster maps | |
| --- | --- |
| Name | <taxon name> |
| CRS | EPSG:3034 - ETRS89-extended / LCC Europe |
| Extent | 1993992.0, 449652.0 : 6843992.0, 5224652.0 |
| Unit | meters |
| Width | 194 |
| Height | 191 |
| Pixel size | 25000, -25000 |
| Data type | Float32 |
| GDAL Driver Description | GTiff |
| File format | GeoTIFF |
| Band count | 3 |
| Band 1 | Occurrence probability (Kriging mean) |
| Band 2 | Occurrence map (<= probability thresholds) |
| Band 3 | Occurrence uncertainty (Kriging variance) |

**Table 2.** Main metadata of the pollen presence point datasets.

| Point data | |
| --- | --- |
| Name | <taxon name> |
| File format | ESRI Shapefile |
| Geometry | Point (Point) |
| CRS | EPSG:3034 - ETRS89-extended / LCC Europe |
| Unit | meters |
| Feature count | 5362 |
| Attribute Count | 1 |
| POLLEN_PRE | String (T=True, F=False) |

## 3.4 How to read the maps

In this section, we illustrate the results of the interpolation framework by introducing the atlas figure content with three common European trees, which are representative of the map diversity of the atlas. Figure 1 shows the results for the taxon *Abies* (fir), which is mainly observed in western and central Europe, around the Black Sea region and in northwestern Russia. The source dataset plotted over the occurrence probability map (Figure 1 a) indicates that the observations match well the high (yellow)

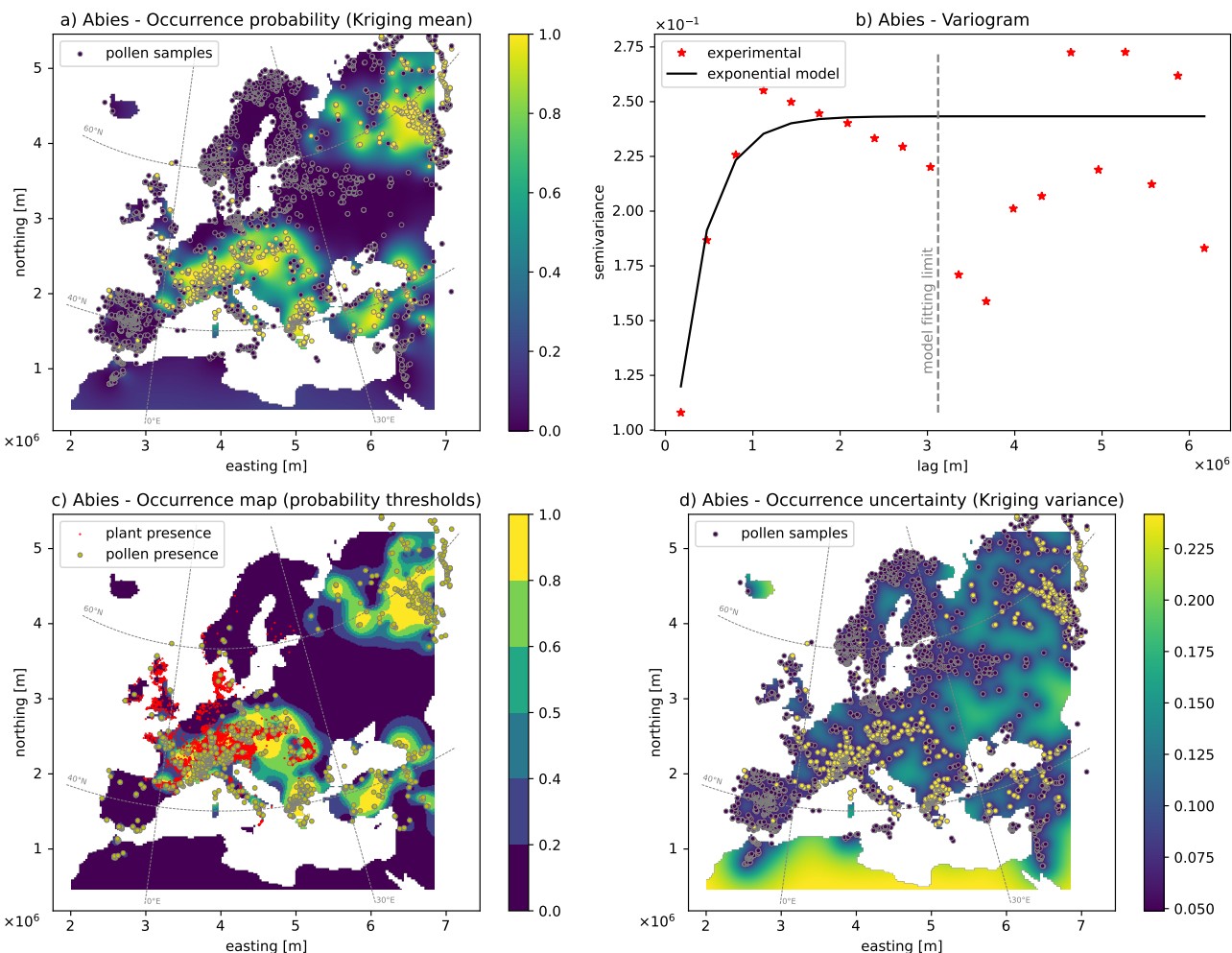

**Figure 1.** Output maps for *Abies*: a) Pollen occurrence probability map, b) Semivariogram model, c) Occurrence map based on probability thresholds, d) Uncertainty map based on the Kriging variance. Red dots in panel c indicate the plant-presence data (see section 3.2.2).

and low (blue) probability areas. Interpolated areas from presence to absence data, or where the two types are densely mixed, present an intermediate probability of pollen occurrence and are accordingly represented by light blue/green shades.

Figure 1 b shows the exponential semivariogram model calibrated with the automatic setup and used in the Kriging interpolation to estimate the occurrence probability map. The semivariogram range is approximately 1000 km, representing the average correlation distance of the data. By imposing thresholds to the probability map, a discrete occurrence map is obtained

(Figure 1 c), delimiting zones related to discrete probability intervals. This version of the probability map is proposed as a ready-to-use tool for practitioners who want to quantify discrete areas of pollen presence. Depending on the application and taxa abundance, different thresholds may be considered.

Frequent *Abies* pollen presence is represented by yellow and green patches covering large parts of the Mediterranean countries, central and eastern Europe, and northwestern Russia. This distribution partly matches the plant distribution data from the external dataset (red dots in Figure 1 c). The high presence of *Abies* in the UK and Denmark, as suggested by the plant distribution data, is not represented in the pollen and is consequently not present in the Kriging interpolation. These plant observations generally constitute introduced trees and might have been excluded from the pollen analysts who generated the data on the basis that they are anthropogenic indicators. The apparent mismatches in northwestern Russia and the Black Sea region are the result of the limited eastward extension of the plant dataset.

Finally, Fig. 1 d shows the Kriging variance map that provides information on the uncertainty of the interpolation. This indicates the variability of the interpolation determined by the distance from the available data, their variability, and the chosen semivariogram model. At the bottom of the map, the lack of data sensibly increases the uncertainty of the pollen presence probability estimates. This is inherent to the data location and common to all taxa of the atlas (See Fig. 4 a).

The second example showcases the pollen distribution of *Betula* (birch, Figure 2), a tree commonly observed north of 45°N. The semivariogram model is fitted with the experimental semivariogram (red stars) in the lower-lag portion used for calibration (below 3100 km, see section 2.1). Compared to the previous example (Figure 1), the semivariogram model for *Betula* shows a larger range around 3000 km. This allows longer correlation structures, necessary to model the extensive pollen presence across Europe (Figure 2 a, c), in agreement with the plant distribution data (red dots, Figure 2 c). In the central-eastern part, the lower density of data moderately increases the model uncertainty as seen by the Kriging variance map (Figure 2 d).

The third example is based on the distribution of the pollen of *Olea* (the olive tree, Figure 3), which is commonly observed across most of the Mediterranean region and disappears rapidly with increasing distance from it. Similarly to the *Betula* case, the exponential semivariogram model presents a large range (3000 km), which accounts for long correlated east-west structures covering the southern sector of the map, where the pollen presence is highly probable (Figure 3 a). Towards the mid-latitudes of Europe, the density of detected pollen points decreases progressively until total absence. Since the spatial structure of the occurrence probability follows a simple north-south gradient, the uncertainty of this map (Figure 3 d) is low and uniform, except at the southernmost edge of the map and in Iceland, where the lack of data increases the uncertainty.

## 3.5   Ensemble reliability assessment

To asses the overall uncertainty of the predictive maps ensemble, we derived a map of the average Kriging variance, here defined as the mean of all the taxa variance maps (Band 3 in Table 1). With the variance theoretically ranging in this case between 0 and 1, the map presents low values in the order of 10e-2 (Figure 4 a), with no zones of high uncertainty over the European continent and its variability controlled by the distance from the data. This confirms that the selected data provide a statistically accurate information on the pollen distribution over the study zone. The poorly constrained regions unsurprisingly lie in data poor regions.

To assess the reliability of the probabilistic predictions, we generated reliability plots, which are realized by removing 50% of the data and then plotting the predicted probability of pollen occurrence for these locations with the observed frequency from the removed data (details in section 2.3). While the amount of removed data is rather high and may penalize the prediction

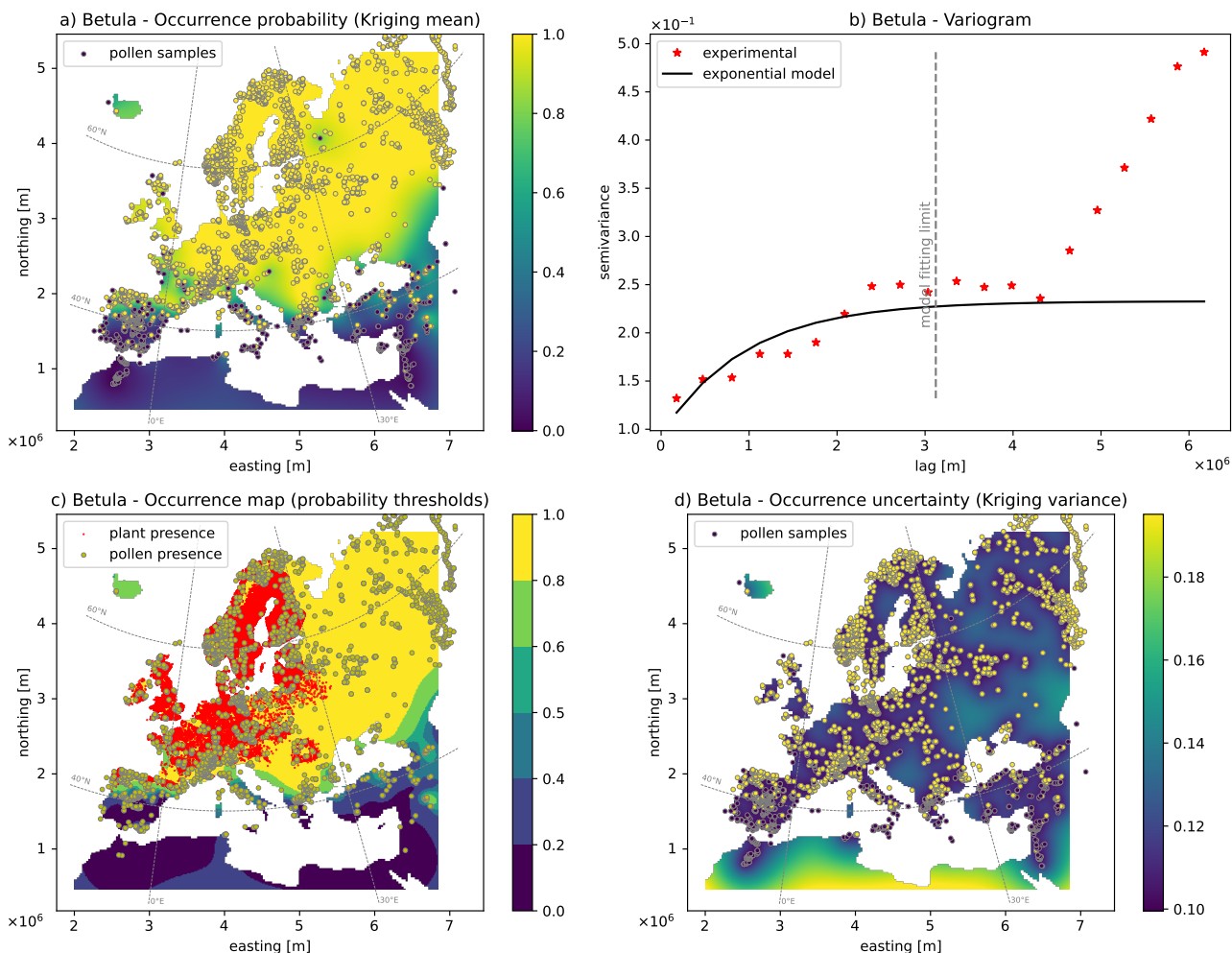

**Figure 2.** Output maps for *Betula*: a) Pollen occurrence probability map, b) Semivariogram model, c) Occurrence map based on probability thresholds, d) Uncertainty map based on the Kriging variance. Red areas in panel c indicate the plant-presence data (see section 3.2.2).

performance, it is necessary to select enough pollen occurrences for all predicted probability classes (see section 2.3). The reliability values are displayed in both form of an ensemble graph (Fig. 4 b) and a table containing the same reliability indicator for each taxon individually (supplemental 1). The latter allows identifying the taxa that do not show reliable predictions for any probability class.

In the ensemble reliability plot (Figure 4 b), the taxa distribution mainly aligns with the bisector (.25-.75 quantile envelope of the ensemble), meaning that the predicted occurrence probability matches well the observed frequency. Nevertheless, for some taxa, the interpolation has a tendency to understimate the pollen occurrence, as seen from the 0.1 ensemble boundary below the bisector (Figure 4 b). This tendency can be identified in the table presented in the supplemental material 1, where the

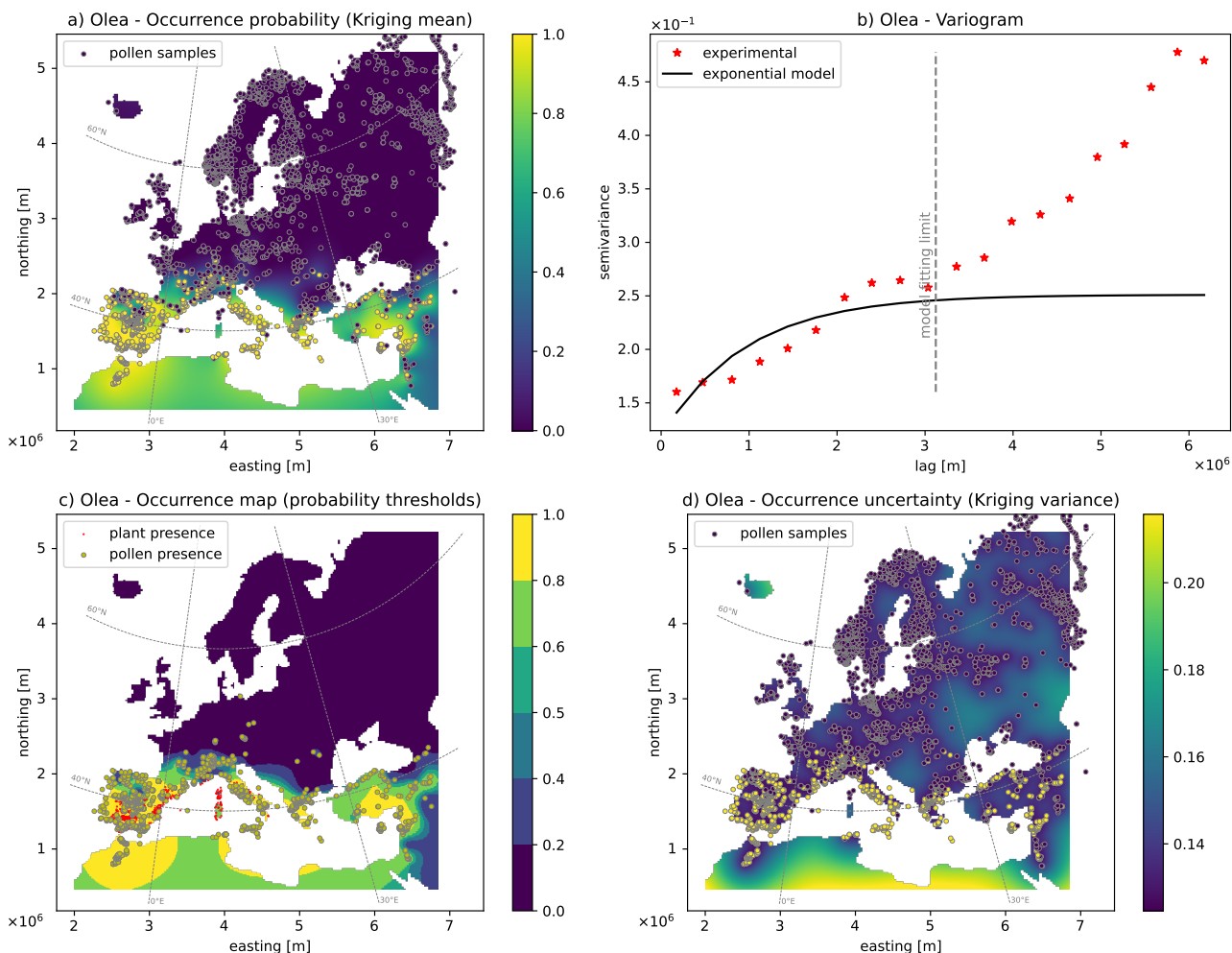

**Figure 3.** Output maps for *Olea*: a) Pollen occurrence probability map, b) Semivariogram model, c) Occurrence map based on probability thresholds, d) Uncertainty map based on the Kriging variance. Red areas in panel c indicate the plant-presence data (see section 3.2.2).

observed occurrence probability in each class is shown for each taxon separately, with the biased values marked in bold. The biased values mainly correspond to taxa with poorly represented local pollen variability, where isolated pollen presence data are surrounded by non-detection points or vice versa. Examples of these cases include *Casuarina*, *Poaceae*, *Tamarix*, and taxa which are almost totally absent as *Aizoaceae*, *Styrax*, *Tsuga*, or *Vitex*. In general, the model is reliable, including for many rare taxa (*i.e.* the ones with only the lowest-probability column filled in supplemental material 1).

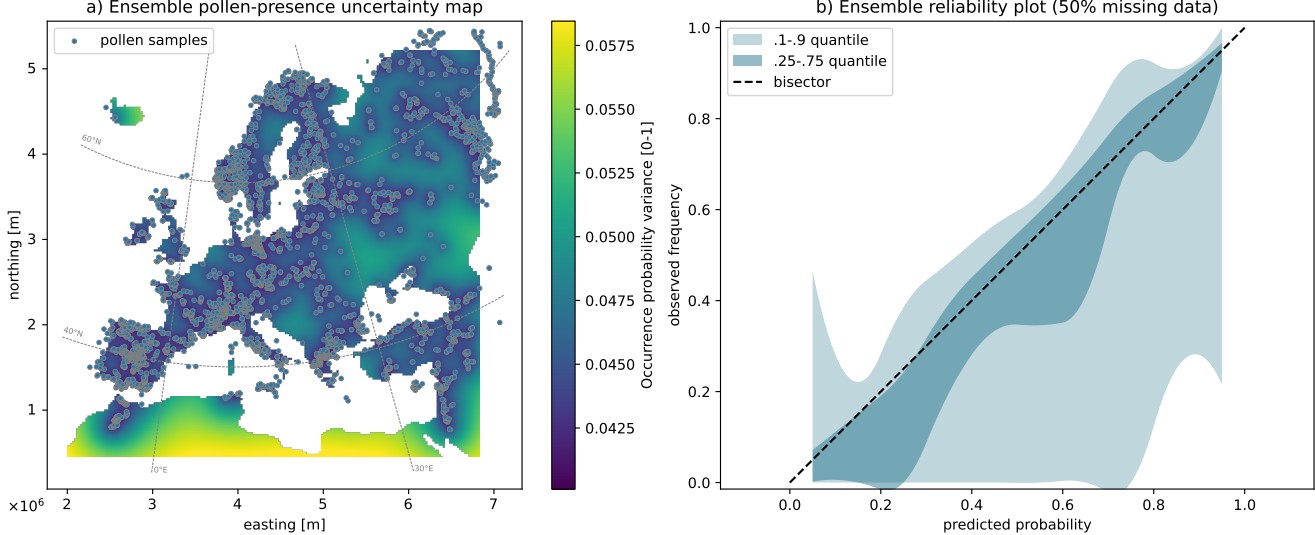

**Figure 4.** Ensemble statistical indicators for the generated maps: a) Average Kriging Variance map and b) Reliability plot obtained from the cross-validation test removing 50% of the data. The latter included the observed occurrence probability (y axis) for all taxa shows as a distribution for every predicted probability class (x axis).

## 4 Conclusions and Perspectives

The presented atlas constitutes a systematic cartographic product offering both a discrete and probabilistic estimation for pollen presence in Europe. A primary application are paleoclimate and paleoenvironmental reconstructions, where these maps can be used as contemporary analogs, but also biodiversity and environmental studies requiring spatially continuous pollen maps as input.

The performed cross-validation test (section 3.5) with 50% of the data removed shows that the interpolation approach is overall reliable and accurate for complex but well represented spatial pollen distributions. The results suggest this is also true for rarely detected taxa. Nevertheless, the data should be sufficient to represent complex local variability or sporadic pollen presence in order to lead to a robust interpolation.

One possibility to relax this requirement could be to integrate auxiliary variables to guide the interpolation, especially for zones where the point data are scarce, *e.g.* by applying the Universal Kriging with external drift. If the auxiliary variable is informative enough, it increases the predictive power of the model, but at the cost of increased computational burden, especially in regional studies with large interpolation grids like the one presented here. However, the improvement of the model by adding this additional information layer is not guaranteed. Indeed, our preliminary attempt to incorporate the elevation variable over the whole interpolation grid as external drift did not lead to any significant improvement in the interpolation. This can be explained by the fact that, at this regional scale, this auxiliary variable does not have a simple statistical relationship with the target one, so it cannot serve as an optimal explanatory variable. Nevertheless it may be the case in sub-regional contexts, when

a clear and causal correlation between elevation and the target variable can be observed. For this reason, an accurate correlation study would be necessary to set up multivariate interpolations and improve our model.

The developed workflow is adaptive to large datasets so it is suitable for regional gridded interpolations. In particular, it should perform equally well with fossil pollen records to produce continuous pollen/vegetation maps during key periods of the past, provided that the data density remains sufficient.

## 5 Code and data availability

Dataset name: EUPollMap Version: 1.1

Release date: 10.2023

Developer: Fabio Oriani

Format: ESRI GeoTIFF, ERSI Shapefile

Repository: https://doi.org/10.5281/zenodo.10015695

Scripts to generate the dataset: https://github.com/orianif/EUPollMap_scripts

License: CC v.4.0

*Author contributions.* Fabio Oriani - conceptual design, development, data analysis, manuscript writing, manuscript revision; Manuel Chevalier - conceptual design, funding acquisition, development, manuscript revision; Gregoire Mariethoz - conceptual design, manuscript revision.

*Competing interests.* No competing interests are present.

*Acknowledgements.* The present study is funded by the Swiss National Science Foundation (SNSF), project CRSK-2 195875. MC is also supported by the German Federal Ministry of Education and Research (BMBF) as a Research for Sustainability initiative (FONA; https://www.fona.de/en, last access: 25 June 2022) through the PalMod Phase II project (grant no. FKZ: 01LP1926D).

We thank very much Dr. Denis Allard, research director at BioSP INRAE, for the helpful contribution as internal reviewer on the modeling approach.

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
