# Peer review of "EUPollMap: The European atlas of contemporary pollen distribution maps derived from an integrated Kriging interpolation approach"

_Earth System Science Data, 2022_

## Author Response (AR1)

Dear Editorial Board,

We thank very much the editors and the reviewers for receiving positively our contribution. We have carefully examined the useful suggestions to which we answer in the following. Indicated line numbers refer to the attached change-tracked version of the manuscript.

In addition to the edits suggested by the reviewers, in the revised version of the dataset we will exclude the use of Gaussian and power models (including the linear one) from the pool of possible variogram models since they are not authorized in Indicator Kriging (Chilès et Delfiner, 2012 p.102). We haven't observed any dramatic changes in the output maps and we have edited the methods (section 2) accordingly.

Reference:

Chiles, Jean-Paul, and Pierre Delfiner. *Geostatistics: modeling spatial uncertainty*. Vol. 713. John Wiley & Sons, 2012.

**RC1**: https://doi.org/10.5194/essd-2022-364-RC1

This paper introduces a Kriging interpolation method aimed at generating comprehensive pollen presence maps for the Euro-Mediterranean Basin. The subject matter of this paper holds considerable interest and is undoubtedly of value to the journal's readership. The paper is generally well-structured, providing all necessary information effectively. I particularly commend the authors for their clear and accessible explanation of the Kriging technique, which aids readers unfamiliar with this field.

While I don't have specific comments on the paper, I do have a couple of suggestions for the authors to consider:

1. In lines 159-162, the authors make an assumption that "any sample location where a taxon has not been observed corresponds to an absence datum for that taxon." It would be beneficial to emphasize the implications of this assumption and its support in the existing literature.

*We thank the reviewer for the valuable suggestion, the following paragraph is added at line 181:*

*Determining the proper absence of a pollen taxon can only be done with extensive vegetation surveys, which is unpractical at the European scale. Moreover, such surveys cannot be done for fossil observations. Therefore, we chose to analyze the EMPD2 dataset as we would analyze the fossil records. For taxa that produce large quantities of pollen grains (grasses, pines), low percentages usually represent long-distance transport to the surroundings of the collection site, without the actual taxon presence [Lisitsyna et al. 2011]. Assuming that their non-observation is proof of absence is therefore reasonable. On the other hand, rare taxa or low-pollinating taxa are more difficult to observe in both modern and fossil settings. It is common to observe them in one sample and not in the neighboring one. Using Kriging at the regional target scale for this study, this problem is mitigated since the presence is assessed as a continuous probability variable, computed as a weighted mean from multiple neighbor presence/absence data.*

*Reference:*

*Olga V. Lisitsyna, Thomas Giesecke, Sheila Hicks, Exploring pollen percentage threshold values as an indication for the regional presence of major European trees, Review of Palaeobotany and Palynology, Volume 166, Issues 3–4, 2011, Pages 311-324, https://doi.org/10.1016/j.revpalbo.2011.06.004.*

2. In lines 195-196, the sentence reads, "The variogram range is approximately 1000 km, corresponding to the variogram range and representing the average correlation distance of the data.". This sentence could be rephrased for clarity and repetitions.

*We thank the reviewer for spotting the error, the sentence will be rephrased as (l.233) "The variogram range is approximately 1000 km, representing the average correlation distance of the data."*

**RC2**: https://doi.org/10.5194/essd-2022-364-RC2

The authors created EUPollMap - pollen occurrence probability maps of 194 pollen taxa for Europe at 25 km spatial resolution using Indicator (Ordinary) Kriging as an spatial interpolation method. The dataset is a valuable source and can serve for various analysis and as input for modeling of various environmental variables. The manuscript is well-organized and well describes the dataset, methodology and validation. I acknowledge the authors for making this research reproducible by opening the code and the data in standard GeoTiff format.

I have one major recommendation on dataset validation and several minor comments/recommendations for the authors.

Major comment:

At first, I didn't quite understand how you validated the dataset from Section 2.3 Validation strategy. I recommend you to better explain how you validated the data. Later on, I understood that you removed 50% of the data so that the predictions and validation set are within the same range and validated the maps for 10 bins (0-0.09, 0.10-0.19, …). I'm generally wondering why you haven't used e.g. stratified 10-fold cross validation for each of the 194 variables? If you take stratified samples you will not have any problem with splitting the data. I think that won't take you much time and it would be good to know the accuracy (R2, RMSE, …) for each of the variables for someone who wants to use the data. You can also analyze which variable has higher or lower accuracy and why.

*We thank the reviewer for bringing this point up and we agree further clarification is needed. Stratified sampling and canonical error scores are not applicable here since the target continuous variable (the probability of pollen occurrence, divided into 10 bins in the reliability plot) is derived from a primary variable, which is the binary (0/1) pollen presence. So, in this case, we cannot stratify in 10 bins the data sampling because data only have 2 possible values (0/1). Error scores such as R2 or RMSE suffer*

*from the same limitation. In this case, it would only be possible to apply them to the threshold maps derived by the probability-of-occurrence field, then compared with 0/1 data. The error would not be a robust metric, since it would be the distance from arbitrary thresholds (see original manuscript lines 196-199). Considering multiple thresholds, this metric would become similar to the reliability plot, with the difference that the latter is a real probabilistic metric. That is why, after considering these possible solutions and previous study cases (l.146-149), we think that the reliability plot is more appropriate for validation in this case.*

*We expanded the validation section 2.3 to better explain the test applied. The paragraph at lines 157-162 will be rephrased with the following: "The way binary presence/absence data aggregate in space determines the estimated probability of occurrence. Some values are rarely found in the output probability map, therefore a high amount of validation data is needed for the reliability plot to be representative of all probability bins. Moreover, the sampling for these data cannot be stratified according to the posterior probability values, which are not available a priori. To cope with this limitation, we randomly removed 50\% of the data to approach stable statistical values for all bins of the reliability plot."*

Minor comments:

We thank the reviewer for the suggestions below, all of them is considered positively and will be applied, see more details where needed.

*We thank the reviewer for the suggestions below, all of them are considered positively and will be applied, see more details below.*

Line 66: I suggest replacing "variogram" to "semivariogram". Most researchers use the word "variogram", but in Kriging we actually use semivariogram (based on semi-variance).

*We agree, the word is replaced everywhere.*

Line 68: change "at coordinates x" to "at spatial locations x" - if these were coordinates, you will have (x, y).

*Agreed.*

Line 70: change "any known values" to "any pair of observations"

*Agreed.*

Line 74: "using a least-square approach" - there are several more approaches to fit the semivariogram, but this one is most common. So change to "mostly using a least-square approach".

*OK, the sentence is edited accordingly.*

Line 76: change "suitable for most situations" to "suitable for the spatially correlated data: …"

*Agreed.*

Line 80: "a smooth variation among adjacent data (zero lag)" - I wouldn't say that because the Kriging prediction is always "smooth". It actually means that there is no measurement error and the prediction error at observation locations will be zero.

*Agreed, the sentence is rephrased as "indicates that the variation among adjacent observations (zero lag) is close to zero".*

Line 80: "intersection the y-axis at positive values" - it's always positive (check the experimental semivariogram eq.).

*Agreed, the sentence is corrected to "intersection at values larger than zero."*

Line 81: "indicates continuities in adjacent data, a phenomenon called nugget effect." - this is actually not true. Nugget represents the variation between the observations at short distances (it can be seen as the measurement error or observational dataset variation). If you have some value for the nugget, it mustn't always be the nugget effect. You can have a large nugget but there is still spatial correlation between the observations. The nugget effect is when you have nugget and there is no visible spatial correlation between the observations (you cannot fit any of the mathematical models you mentioned in Line 77). Please rephrase this.

*We agree, the sentence is rephrased as "indicates the variation in adjacent data".*

Lines 88 and 90: Give some reference for OK and UK. When you say OK, you mean Indicator Kriging - mention this also

*The OK and UK acronyms are defined when firstly mentioned at line 98. OK can be applied to indicator sets, as explained at lines 106-110.*

Line 91: "No sensible differences in the resulting interpolations were observed" - explain this. Is it because that there is no correlation with the elevation (you can give some accuracy metric, e.g. R2 from the linear regression) or it can be correlation, but there is not much spatial correlation left between the residuals. - I see now that you explained this in the Conclusions and Perspectives section, but still recommend that you explain it here too.

*We agree, the explanation is reported earlier as suggested (l.102-105)*

Line 91: "OK allowed including all data" - what do you mean by this? Is there no elevation for all observation locations?

*We agree, the sentence is rephrased (l.102-105) "OK allowed including all data points with reasonable computation time, while UK required an excessive computational burden. Also, elevation did not correlate with the pollen presence in the analyzed data, so that its inclusion did not sensibly affect the prediction when used as external drift in the Kriging model. For these reasons OK was preferred over UK.".*

Line 98: "and indicates the uncertainty in space of the estimated probability" - the Kriging variance is actually a metric that shows the estimation precision at a specific location and is dependent on the number of samples and spatial distribution of the samples, and is independent of the values of the samples. Please rephrase this.

*We agree, the explanation of the variance field should be clarified. Variance independence from the data values is true, for a given covariance model (Goovaerts 1997, p. 179), while in this and most practical cases, the variogram model is fitted on the data. As a consequence, the OK variance field will also depend indirectly on the data values. We obtain different variance fields for different pollen species, presenting the same data amount and locations, only changing their 0/1 values. We rephrase the sentence as (l.111) "and indicates the uncertainty of the prediction depending on the data amount, their spatial distribution, and the semivariogram model."*

*Reference:*

*Goovaerts, P., 1997, Geostatistics for Natural Resources Evaluation, Oxford University Press.*

Line 104: add "experimental semivariogram"

*Agreed.*

Line 106: change "e.g. with" to "such as"

*Agreed.*

Line 111: "impose flat or monotonic-positive model functions" - explain why - because we expect this if there is spatial correlation

*We agree, this sentence is modified as (l.126-129) "Monotonicity and a positive slope are expected in a model semivariogram, but, in case of a noisy experimental semivariogram (which is the case for some of the observed pollen taxa), unconstrained fitting can lead to a negative slope in the model. For this reason, we impose flat or monotonic-positive model functions ".*

Line 114: For step one, were there any cases when there was no spatial correlation? If so, how did you handle this?

*We agree, we added the sentence at line 135: "When the data have little to no spatial correlation, the fitted semivariogram model tends to become a constant function, which subsequently leads to constant estimated mean and variance fields."*

Line 118: "Solve the Kriging system" - change to "Kriging prediction"

*Agreed.*

Line 124: "visually inspected" - how? I suppose you couldn't do this for all 194 maps.

*We agree, we corrected to "We constituted an example dataset by identifying a series of common species for Europe with characteristic spatial features (e.g., broad extent, rare species, discontinuous distributions). The interpolated probability surfaces and their variances were then visually inspected and compared with the original observations".*

Line 127: "removing part" - say here that you split the data on train and test dataset.

*We agree.*

Line 227: "as the mean of all taxa variance maps" - you take the average, bu you can also take min and max to check where and for which variable the models perform best and worst (see the major comment , the cross validation for each variable)

*We agree, see next answer.*

Line 237: You refer here to supplemental 1 with the accuracy for each of the variables - you can mention which one performs the best, which one the worst and why…

*We haven't detected any particular taxonomic group showing a worst/best performance, while we have observed that the worst cases are the ones showing a complex spatial heterogeneity, with isolated presence/absence points. Additional examples have been added at lines 287-290.*